# Mortality Predictive Value of the C_2_HEST Score in Elderly Subjects with COVID-19—A Subanalysis of the COLOS Study

**DOI:** 10.3390/jcm11040992

**Published:** 2022-02-14

**Authors:** Piotr Rola, Adrian Doroszko, Małgorzata Trocha, Katarzyna Giniewicz, Krzysztof Kujawa, Marek Skarupski, Jakub Gawryś, Tomasz Matys, Ewa Szahidewicz-Krupska, Damian Gajecki, Barbara Adamik, Krzysztof Kaliszewski, Katarzyna Kilis-Pstrusinska, Krzysztof Letachowicz, Agnieszka Matera-Witkiewicz, Michał Pomorski, Marcin Protasiewicz, Konrad Majchrzak, Janusz Sokołowski, Ewa Anita Jankowska, Katarzyna Madziarska

**Affiliations:** 1Department of Cardiology, Provincial Specialized Hospital, Iwaszkiewicza 5 Str., 59-220 Legnica, Poland; 2Clinical Department of Internal Medicine, Hypertension and Clinical Oncology, Wroclaw Medical University, Borowska 213, 50-556 Wroclaw, Poland; adrian.doroszko@umw.edu.pl (A.D.); jakub.gawrys@umw.edu.pl (J.G.); tomasz.matys@umw.edu.pl (T.M.); ewa.szahidewicz-krupska@umw.edu.pl (E.S.-K.); damian.gajecki@umw.edu.pl (D.G.); 3Department of Pharmacology, Wroclaw Medical University, Mikulicz-Radecki Street 2, 50-345 Wroclaw, Poland; malgorzata.trocha@umw.edu.pl; 4Statistical Analysis Centre, Wroclaw Medical University, K. Marcinkowski Street 2-6, 50-368 Wroclaw, Poland; katarzyna.giniewicz@umw.edu.pl (K.G.); krzysztof.kujawa@umw.edu.pl (K.K.); 5Faculty of Pure and Applied Mathematics, Wroclaw University of Science and Technology, Wybrzeże Wyspiańskiego Street 27, 50-370 Wroclaw, Poland; marek.skarupski@pwr.edu.pl; 6Clinical Department of Anaesthesiology and Intensive Therapy, Wroclaw Medical University, Borowska Street 213, 50-556 Wroclaw, Poland; barbara.adamik@umw.edu.pl; 7Department of General, Minimally Invasive and Endocrine Surgery, Wroclaw Medical University, Borowska Street 213, 50-556 Wroclaw, Poland; krzysztof.kaliszewski@umw.edu.pl; 8Clinical Department of Paediatric Nephrology, Wroclaw Medical University, Borowska Street 213, 50-556 Wroclaw, Poland; katarzyna.kilis-pstrusinska@umw.edu.pl; 9Clinical Department of Nephrology and Transplantation Medicine, Wroclaw Medical University, Borowska Street 213, 50-556 Wroclaw, Poland; krzysztof.lechtanowicz@umw.edu.pl (K.L.); konrad.majchrzak@gmail.com (K.M.); katarzyna.madziarska@umw.edu.pl (K.M.); 10Screening of Biological Activity Assays and Collection of Biological Material Laboratory, Wroclaw Medical University Biobank, Wroclaw Medical University, Borowska Street 211A, 50-556 Wroclaw, Poland; agnieszka.matera-witkiewicz@umw.edu; 11Clinical Department of Gynecology and Obstetrics, Wroclaw Medical University, Borowska Street 213, 50-556 Wroclaw, Poland; michal.pomorski@umw.edu.pl; 12Clinical Department and Clinic of Cardiology, Wroclaw Medical University, Borowska Street 213, 50-556 Wroclaw, Poland; marcin.protasiewicz@umw.edu.pl; 13Department of Emergency Medicine, Wroclaw Medical University, Borowska Street 213, 50-556 Wroclaw, Poland; janusz.sokolowski@umw.edu.pl; 14Institute of Heart Diseases, Wroclaw Medical University, Borowska Street 213, 50-556 Wroclaw, Poland; ewa.jankowska@umw.edu.pl; 15Institute of Heart Diseases, University Hospital in Wroclaw, Borowska Street 213, 50-556 Wroclaw, Poland

**Keywords:** COVID-19, elderly, C_2_HEST-score, SARS-CoV2, mortality, risk-score, outcomes, senility, predictive value

## Abstract

Senility has been identified among the strongest risk predictors for unfavorable COVID-19-outcome. However, even in the elderly population, the clinical course of infection in individual patients remains unpredictable. Hence, there is an urgent need for developing a simple tool predicting adverse COVID-19-outcomes. We assumed that the C2HEST-score could predict unfavorable clinical outcomes in the elderly subjects with COVID-19-subjects. Methods: We retrospectively analyzed 1047 medical records of patients at age > 65 years, hospitalized at the medical university center due to COVID-19. Subsequently, patients were divided into three categories depending on their C2HEST-score result. Results: We noticed significant differences in the *in-hospital* and 3-*month* and 6-*month* mortality-which was the highest in *high*-*risk*-C2HEST-stratum reaching 35.7%, 54.4%, and 65.9%, respectively. The *medium-risk*-stratum mortalities reached 24.1% 43.4%, and 57.6% and for *low-risk*-stratum 14.4%, 25.8%, and 39.2% respectively. In the C2HEST-score model, a change from the *low* to the *medium* category increased the probability of death intensity approximately two-times. Subsequently, transfer from the *low-risk* to the *high-risk*-stratum raised all-cause-death-intensity 2.7-times. Analysis of the secondary outcomes revealed that the C2HEST-score has predictive value for acute kidney injury, acute heart failure, and cardiogenic shock. Conclusions: C2HEST-score analysis on admission to the hospital may predict the mortality, acute kidney injury, and acute heart failure in elderly subjects with COVID-19.

## 1. Introduction

The novel severe acute respiratory syndrome coronavirus 2 (SARS-CoV2) causing Coronavirus disease 2019 (COVID-19), firstly described as a local cluster of pneumonia in Wuhan, Hubei, China [1], despite initial widespread use of preventive measures for personal protection [2], has spread worldwide and evolved into a global pandemic, affecting healthcare systems all over the world.

Although many risk factors for the disease progression have been identified, clinical course of infection in individual patients remains still uncertain. Senility, male gender, obesity, previously diagnosed cardiovascular disorders, diabetes, and chronic pulmonary diseases, are known as mortality risk factors [3]. Furthermore, various laboratory abnormalities, including immunological, hematological, and biochemical changes along with specific computed tomography findings are postulated [4] to predict the severity of the disease and its outcome. Among the mentioned risk factors, particularly advanced age (over 65 years) is the most prominent risk factor for an unfavorable outcome [5]. Despite the high risk attributed to this subpopulation, clinical experience indicates that the course of COVID-19 is heterogeneous, ranging from asymptomatic to fatal cases. Facing limited resources during COVID-19 pandemic, adequate selection of patients with the highest probability of unfavorable outcome is crucial for designing individualized diagnostic and therapeutic strategy.

The C2HEST-score is a simple, well-established [6] scoring system, allowing stratification of the risk of developing atrial fibrillation (AF). Recently Liang et al. [7] demonstrated that the C_2_HEST score could also predict adverse outcomes including death and hospitalization among patients with heart failure. Considering that the individual components of the C2HEST score are identical to those risk factors attributed to worse clinical course of COVID-19, we assumed that the C2HEST could predict an unfavorable clinical outcome in COVID-19. In this study, based on the data from the COLOS registry, we performed a subanalysis of the elderly population with COVID-19 assessing the diagnostic performance of the C2HEST score for fatal and non-fatal clinical outcomes.

## 2. Materials and Methods

### 2.1. Study Design and Population

In the present study, we described the clinical characteristics of 1047 elderly (over 65 years) Patients with COVID-19 hospitalized at the University Hospital in Wroclaw between February 2020 and June 2021. All medical records were collected as part of the COronavirus in Lower Silesia—the COLOS registry. Subjects chosen to this study were retrospectively selected out of all (2184) COLOS study participants. The sole inclusion criterion to this subanalysis was age of >65 years in the Patients with COVID-19. There were no other additional exclusion criteria regarding, patient’s clinical characteristic, comorbidities nor severity of COVID-19. Figure 1 presents study protocol. 

The initial diagnosis of SARS-CoV2 was confirmed with reverse transcription–polymerase chain reaction (RT-PCR) for viral RNA of nasopharyngeal swab specimens.

The COLOS study protocol has been approved by the Institutional Review Board and Ethics Committee at the Wroclaw Medical University, Wroclaw, Poland (No: KB-444/2021). The written informed consent to participate was waived due to the retrospective, observational nature of the study. The Bioethics Committee approved the publication of fully anonymized data.

### 2.2. Clinical Follow-Up and Outcomes

All the study participants underwent clinical assessment during the hospital admission. Past medical history, home medication, and vital parameters were assessed in every subject. Similarly, initial blood samples were drawn in every patient at the time of hospital admission, during the course of hospitalization and at discharge time. Clinical follow-up included the whole in-hospital period and ended on the day of discharge or death. In the post-discharge period, data regarding death were collected up to 6 months.

The primary outcomes included: in-hospital mortality, 3-month and 6-month all-cause mortality, the end of hospitalization other than due to death (discharge home/emergency transfer to another center–deterioration/transfer for rehabilitation). Secondary outcomes included: the need for mechanical ventilation support, myocardial injury, shock, acute heart failure, pulmonary embolism, stroke, acute kidney injury, acute liver dysfunction, pneumonia, sepsis, systemic inflammatory response syndrome (SIRS), multiple organ dysfunction syndrome (MODS), and bleedings.

### 2.3. Study Groups

Patients were assigned to one out of the three arms depending on their C2HEST score result calculated on the hospital admission. Six variables, including coronary artery disease (1 point), chronic obstructive pulmonary disease (1 point), hypertension (1 point), elderly (age ≥ 75 years, 2 points), systolic HF (2 points), and thyroid disease (1 point) were taken into account and defined basing on the patient past medical history and interview at the time of admission. Moreover, in subsequent sensitivity analyses, the “thyroid disease” was replaced more precisely with “hyperthyroidism” and “hypothyroidism”.

After calculating the C2HEST score, patients were allocated to the separate groups depending on the result: –the *low-risk* of 0 to 1 point, –the *medium-risk* of 2 to 3 points, –the *high-risk* of ≥ 4 points.

### 2.4. Statistical Analysis 

Descriptive data are presented as numbers and percentages for categorical variables, and as mean with standard deviation range (minimum–maximum) and number of non-missing values for numerical variables. As omnibus test chi-square test was used for categorical variables with more than 5 expected cases in each group, whereas Fisher exact test was used for cases with fewer cell counts. Welch’s ANOVA was performed for continuous variables due to unequal variances between risk-strata and sample size large enough for appropriateness of asymptotic results. Post-hoc analysis for continuous variables was performed using the Games–Howell test with Tukey correction. For categorical variables, post-hoc test was the same as the omnibus test, but performed in subgroups with Bonferroni correction.

In-hospital mortality and all-cause mortality data were available as right-censored data, thus time-dependent ROC analysis with Inverse Probability of Censoring Weighting (IPCW) estimation was performed for those variables. The C_2_HEST score was assessed through the time dependent area under the curve (AUC). Log-rank test was used to confirm differences in survival curves between risk strata. Proportional hazard assumption was verified using Grambsch–Therneau test. A Cox proportional hazard model was used to analyze the hazard ratio (HR) for the C_2_HEST score, its components, and risk strata.

For secondary outcomes, due to their dichotomic nature, a logistic regression model was fitted. Classical ROC analysis was performed, and AUC measure was used for assessing predictive capabilities. Odds ratio (OR) was reported as effect size for influence of the C_2_HEST score, its components, and risk strata. 

All statistical analyses were performed with R version 4.0.4 using packages time–ROC, pROC [8], survival [9], coin [10], and odds ratio [11]. A significance level of 0.05 was selected for all statistical analyses.

## 3. Results

### 3.1. Patients Baseline Characteristics

Baseline Patient Characteristics are summarized in the Table 1. The *medium-risk* group was the most numerous (419 subjects) and most of the patients in this group were female. Patients in the *high-risk* stratum were older, when compared to other groups. These patients had also the highest prevalence of comorbidities including hypertension, diabetes (DM), dyslipidemia, atrial fibrillation (AF), previous myocardial infract (MI) and percutaneous coronary intervention (PCI), valvular heart diseases, stroke, chronic obstructive pulmonary disease (COPD), heart failure (HF), chronic kidney diseases (CKD), and peripheral artery disease (PAD) history.

Due to the higher prevalence of cardiovascular disease in the *high-risk* stratum, we observed differences in the treatment applied before hospitalization. Subjects in this group more frequently received angiotensin-converting-enzyme inhibitors (ACEI), mineralocorticoid receptor antagonists (MRA), b-blockers, diuretics, statins, vitamin K antagonists (VKA), new oral anticoagulants (NOAC), acetylsalicylic acid, the P_2_Y_12_ inhibitors, and insulin. On the other hand, patients in the *low-risk* group more often were given immunosuppressants other than oral corticosteroid. All the data regarding treatment applied before hospitalization is shown in Table 2.

The high-risk group had a significantly higher prevalence of dyspnea with rales, wheezing, pulmonary congestion, and peripheral edema on admission. No other significant differences in prevalence of other symptoms among the three C_2_HEST risk strata were observed. Noteworthy, there were no differences regarding the Vulnerable Elderly Survey (VES-13) nor the Glasgow Coma Scale (GCS) on admission. All patient-reported symptoms, vital signs, and abnormalities measured during a physical examination at hospital admission are summarized in the Table 3.

### 3.2. Laboratory Assays

The initial laboratory parameters as well as those measured at the end of hospitalization are pooled in the Table 4. At admission, the *high-risk* group was characterized by the lowest level of haemoglobin and blood platelet count. At the same time, this cohort had a significantly higher potassium ion concertation with coexisting elevated INR. Similar observation was made for the renal function parameters. In the *high-risk* group, we observed higher serum level of urea and creatine coexisting with lower eGFR and albumin values. Subjects from the *high-risk* stratum had initially highest mean level of cardiac injury biomarkers (BNP, NT-proBNP and troponin). Compared with patients in the *low-risk* stratum, those in the high-risk had higher serum TSH level, but without significant differences regarding the peripheral thyroid hormones.

### 3.3. Drug Therapy and Applied Treatment during Hospitalization

#### 3.3.1. Drug Therapy

Overall, there were no differences among applied treatment during hospitalization between the three C2HEST risk-strata. The only exception was the prevalence of convalescent plasma application. Subjects from the low-risk stratum more often received this therapy. Data regarding the general management of study subjects are presented in the Table 5.

#### 3.3.2. Treatment Procedures

Greater C2HEST score was associated with the more frequent use of catecholamines. On the other hand, patients in the *low-risk* stratum statically more often did not require any respiratory support. Interestingly, we observed a higher prevalence of patients treated with invasive ventilation in this group (Table 6).

### 3.4. Clinical Outcome

#### 3.4.1. Correlation of C2HEST Score Results and Mortality

The data regarding associations between the C2HEST risk stratum and mortality are presented in Table 7. We noticed significant differences regarding in-hospital, then the 3-month and 6-month mortality, which was the highest in high-risk C2HEST stratum reaching 35.7%, 54.4%, and 65.9%, respectively. Noteworthy, in the medium-risk stratum, the mortality rate reached 24.1%, 43.4%, and 57.6%, whereas in the low-risk stratum, it reached 14.4%, 25.8%, and 39.2%, respectively.

The time-depended discriminatory performance of the C2HEST score on all-cause mortality is presented in Figure 2. The time-dependent AUC for the C2HEST score in predicting all-cause mortality in period reaching from the day of hospital admission up to 240 days after the initial diagnosis was above 60.

Figure 3 shows the monthly time-dependent receiver operating characteristics (time–ROC) related to the C2HEST score. During a whole period, C2HEST maintained at similar level, allowing to predict the mortality with AUC ranging from 60.2 to 63.4.

As a next part of the assessment of the C2HEST score performance in predicting all-cause mortality among elderly subjects with COVID-19, survival curves for all C2HEST strata using Kaplan–Meier functions were estimated. The p value for Log-rank test was <0.0001. Figure 4 shows time-depending survival probability for the three risk strata.

Additionally, two Cox models were analyzed to assess the effect of the C2HEST score stratification on COVID-19 mortality. In the overall model for the uncategorized C2HEST score value, the Grambsch–Therneau test rejected the null hypothesis. The confidence intervals and p values were omitted as they might have been unreliable. On the other hand, considering the categorized-model change from the *low* to the *medium* category increased death intensity approximately 2-times. Subsequently, transfer between the *low-risk* stratum to *high**-risk* stratum raised all-cause death intensity 2.7 times. (Table 8.)

The associations of individual C_2_HEST score components with mortality are presented in Table 9. The highest prognostic value for all-cause-death beyond age had coronary artery disease and heart failure components.

Finally, to verify that the adequacy of the original risk stratification (the low/medium/high-risk categories for 0–1/2–3/≥4 points) provides the best possible stratification regarding the difference in Kaplan–Meier survival curves, all the possible C2HEST intervals were analyzed, and for each, the log-rank statistics were calculated (Table 10). The highest value of log-rank test statistics was obtained for the original C2HEST-score risk strata.

#### 3.4.2. Correlation of the C_2_HEST Score with Secondary Outcome

All clinical non-fatal events and hospitalization are shown in Table 11. Patients in the *high-risk* stratum were more likely to develop acute kidney injury, acute heart failure, and cardiogenic shock. Noteworthy, no-significant differences were reported in the occurrence of pneumonia, SEPSIS, systemic inflammatory response syndrome (SIRS), and multi-organ dysfunction syndrome (MODS). Additionally, there were no differences in the ratio of thromboembolic events (deep vein thrombosis, pulmonary embolism). Similarly, an increase in the C2HEST score did not raise the prevalence of total or gastrointestinal bleedings.

Summarized discriminatory performance of the C_2_HEST score on the clinical events is presented in Table 12 and in the Appendix A.

The associations of individual C2HEST score components with endpoints are presented in the Appendix A. Since Obesity and Diabetes mellitus constitute important comorbidities affecting the COVID-19 outcome, we decided to perform a subanalysis including these two parameters to the modified C2HEST score (C2HEST-OD), which has further increased the predictive performance of the score. The data on the C2HEST-OD score is presented in the Appendix A.

## 4. Discussion

Advanced age is considered as an independent predictor of in-hospital mortality in the course of COVID-19 [12]. Combined with comorbidities and frailty, it leads to the increased risk of an unfavorable outcome in this specific population. The high prevalence of atypical symptoms [13] and more rapid progression of disease indicated that the development of a simple risk-scoring system faced with limited resources could optimize the treatment process.

Some elderly subjects can recover spontaneously without any medical intervention when the disease course is mild. However, in severe cases, despite the use of intensive pharmacological therapy, non-invasive and invasive mechanical ventilation, or extracorporeal membrane oxygenation (ECMO), the prognosis remains poor. Therefore, it is crucial to identify potentially severe cases and implement immediately effective treatment to prevent the progression of the disease from its beginning. Interestingly, there were no significant differences on admission in terms of the Vulnerable Elderly Score (VES-13), which is a simple scoring system capable of identifying vulnerable elderly people in the community and includes factors such as age, self-assessed health, functional limitations, and impairments [14]. Health vulnerability is associated with a higher risk of mortality and functional decline in older people in the community. However, few studies have evaluated the role of the Vulnerable Elders Survey (VES-13) in predicting clinical outcomes of hospitalized patients [15,16]. One of the recent studies, based on the small cohort (n=91) suggests that elderly patients (>60 years) classified as extremely vulnerable had more unfavorable outcomes after hospitalization for COVID-19—super vulnerability was an independent predictor of death and the need for invasive mechanical ventilation during hospitalization—a final VES-13 score between 8 and 10 was associated with poor outcomes [17]. Our results show a lack of significant differences in the VES-13 between the three C2HEST strata. Similarly, we did not observe differences in the GCS score between the risk strata, which could point thus at an independent predicting value of the C2HEST score in the fatal and non-fatal outcomes of elderly subjects with COVID-19. In the Appendix A, we have presented the usefulness of the C2HEST score in elderly subjects who were admitted directly to the intensive care unit (due to COVID severity) vs. those admitted to the non-intensive ward of the medical university center due to COVID-19. The C2HEST score revealed to determine the outcome (mortality and non-fatal adverse clinical events) irrespective of the initial symptom severity. Noteworthy, C2HEST score also predicted the mortality irrespective of the transfer to the ICU, which might point at its additional value in better predicting the need for advanced supportive care and performing better triage of subjects being at greater risk for death who could take an advantage of earlier escalation of the monitoring and supportive care.

Since SARS-CoV-2 affects mainly the respiratory system, classic parameters of ventilation (respiratory rate, oxygen saturation, and PaO2/FiO2) are often used in clinical practice to assess the disease severity. Similar, due to the postulated critical role of inflammatory response in severe COVID-19 systematic inflammation factors, CRP, interleukin-6, [18] and interleukin-8 [19], neutrophil-to-lymphocyte ratio [20] are assumed to correlate with clinical outcome.

However, satisfactory methods for predicting the outcome of hospitalized COVID-19 especially in elderly subjects are still missing. Therefore, we conducted this study to assess the predictive value of C2HEST score in elderly (over 65 years) patients with COVID-19. In the past, the C2HEST score was validated as a simple tool for predicting AF in the general [6] and post-stroke [21] population.

Considering that all the variables (coronary artery disease [22]; chronic obstructive pulmonary disease [23]; hypertension [24]; elderly [25]; systolic heart failure [26]; thyroid disease [27]) of the scale are also factors of an unfavorable prognosis among patients with COVID-19, we assumed C2HEST could predict other clinical outcomes in elderly patients with COVID-19.

Initial laboratory parameters seem to support this theory. In our cohort, the high-risk C2HEST stratum had a higher prevalence of renal insufficiency, initial anemia, and elevated markers of heart injury. A notable trend in the initially higher level of inflammatory parameters was observed, albeit without statistical significance. Lack of statistical significance between C2HEST-dependent risk-groups in terms of initial inflammatory markers and primary respiratory parameters allows presuming that this scale not only selects initially extremely severe cases with poor prognosis, but was able to predict the outcome of the COVID-19 infection from all-comers elderly cohort.

The C2HEST score analyzed as categorical variable well correlated with mortality, acute renal and cardiac complications. When calculated, C2HEST scores were grouped into *low*, *intermediate*, and *high-risk* strata; all three categories were associated with significant differences in terms of the in-hospital mortality for each of the study groups. Moreover, this relationship referred also to the three-month and six-month mortality. Furthermore, the log-rank statistics performed in this study confirmed that the original stratum allocation system used in the C2HEST scale provides the best possible model of mortality stratification.

Our data suggest that among all individual CHEST score components, the highest prognostic value for mortality had an age, coronary artery disease, and heart failure. Surprisingly, previously well-established in general population risk-factors COPD and hypertension [24,28] had no effect on the survival curve in the elderly population.

Among other interesting findings of our study were significant differences in the prevalence of respiratory support applied during the hospitalization. Not surprisingly, patients in the *low-risk* stratum statically less frequently required respiratory support. However, at the same time, they were more prone to deteriorate and required invasive ventilation in intensive care unit (ICU). Probably in the face of limited resources, subjects in this stratum, due to lower prevalence of comorbidities, were predisposed to receive this advanced treatment while patients in *high-risk* stratum had not been qualified for that kind of escalated therapy.

Recently, we observed an instantly growing number of risk scores and predictive models designed for a similar purpose. Especially the elder population with co-occurring immunological changes named collectively as “immunosenescence” [29]—connected with a decrease of innate and adaptive immune responses and exacerbation in the production of inflammatory cytokines—during the aging process is susceptible to various infections and requires careful initial assessment. Some of them use advanced mathematical models based on machine learning. The vast majority of these models use the initial laboratory features, along with respiratory parameters as differentiating variables [30,31,32,33] which may reduce their usefulness in common clinical practice. Moreover, introducing novel scales or scoring systems requires detailed validation and is much more difficult to implement to the common clinical use by medical practitioners. As a result, analysis of the usefulness of the pre-existing scales in the other entity may have much further going practical implication, especially while meeting the urgent need during the COVID-19 pandemic. The C2HEST score seems to be one of the few, well-validated, based only on a simple medical history, and can be applied at early stages of hospital admission or even during the pre-hospital triage.

An interesting concept might be also a multidimensional assessment of a potential risk factor of an unfavorable outcome of COVID-19 in the elderly population, a merger of the C2HEST risk score with some basic clinical factor. Since obesity and diabetes constitute important comorbidities which could affect the COVID-19 outcome, including these parameters to this analysis could further improve the prognostic value of such modified C2HEST-OD which is presented in the Appendix A. Nevertheless, as specified above, the validation of the new scale and introducing it to the clinical practice would take much time which is critical in the pandemic setting. Noteworthy, the CHA2DS2Vasc score, commonly used in clinical practice for estimating the risk of stroke in people with atrial fibrillation (AF), includes comorbidities such as diabetes, but also congestive heart failure, hypertension, prior stroke/TIA or thromboembolism, vascular disease (e.g., peripheral artery disease, myocardial infarction, aortic plaque), and sex category. Similar to the C2HEST score, it is well validated and based on the simple analysis of comorbidities. We postulate that the CHA2DS2Vasc score might also have prognostic value in predicting the COVID-19 outcome in elderly subjects, which requires further detailed and extensive analyses. Additional data including laboratory parameters, frailty assessment value, or radiological features could increase the predictive power of the C2HEST score. Such a combined model could allow for the accurate selection of subjects hospitalized with COVID-19 with the urgent need of introducing life-saving intervention. However, this approach may significantly increase complications of the scale, reducing its practical usefulness.

### Limitations

This study has several limitations. First, the retrospective character and a single-center registry could affect clinical outcomes. Secondly, the study covered a relatively long period and was carried out in the face of limited resources, which could affect therapeutic methods. Finally, some clinical data and baseline laboratory assays are incomplete, hindering proper interpretation of the results.

## 5. Conclusions

This is the first presentation that the C_2_HEST score could predict adverse outcomes including the in hospital and six-month-mortality as well as the non-fatal clinical events reflecting deterioration, such as acute kidney injury, acute heart failure, and cardiogenic shock among elderly patients admitted to the hospital with COVID-19. The simplicity of this scale combined with variables based only on the past medical history with omission of laboratory assays allows assuming that the C2HEST score can be a helpful tool for pre-hospital risk stratification in elderly subjects with SARS-CoV-2 infection.

## Figures and Tables

**Figure 1 jcm-11-00992-f001:**
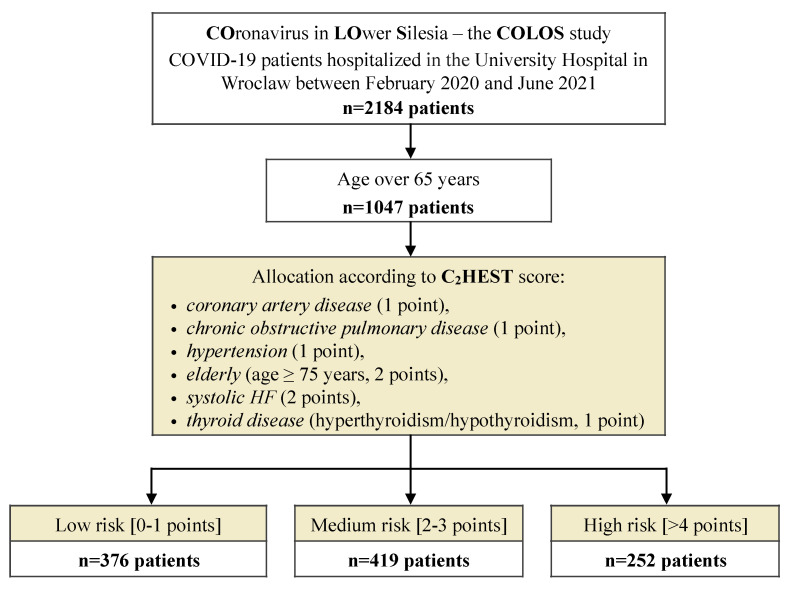
A flow chart presenting the study protocol.

**Figure 2 jcm-11-00992-f002:**
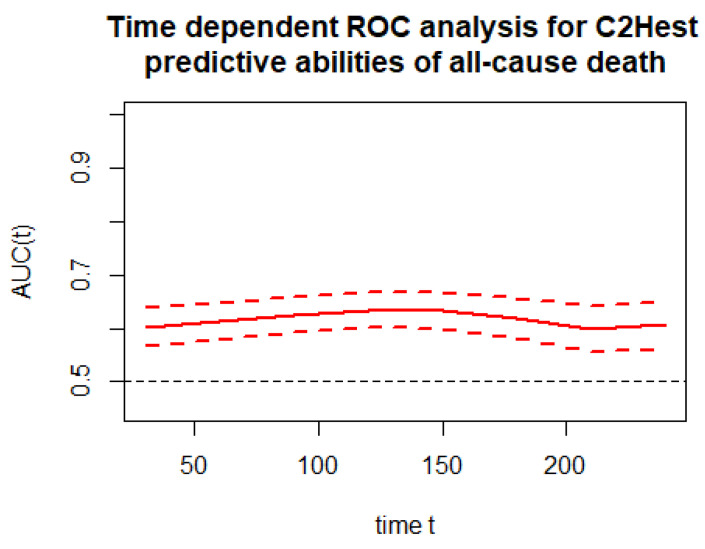
Time-dependent ROC analysis for the C_2_HEST predictive abilities of all-cause death.

**Figure 3 jcm-11-00992-f003:**
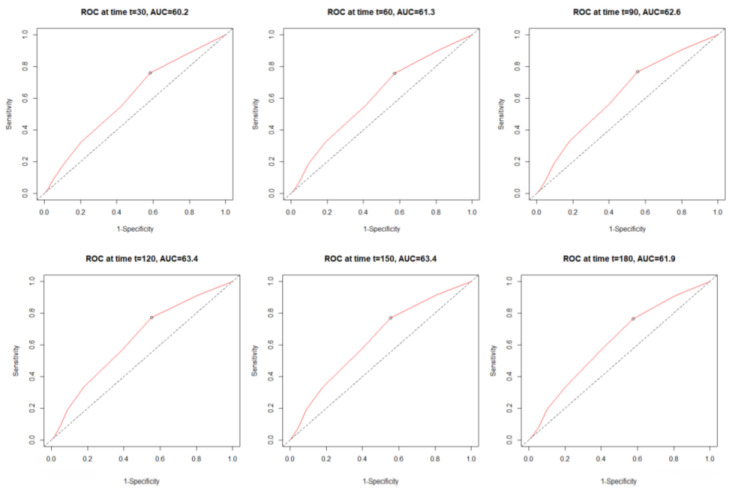
Time-dependent receiver operating characteristic (time–ROC) curves for the C_2_HEST score in predicting total mortality.

**Figure 4 jcm-11-00992-f004:**
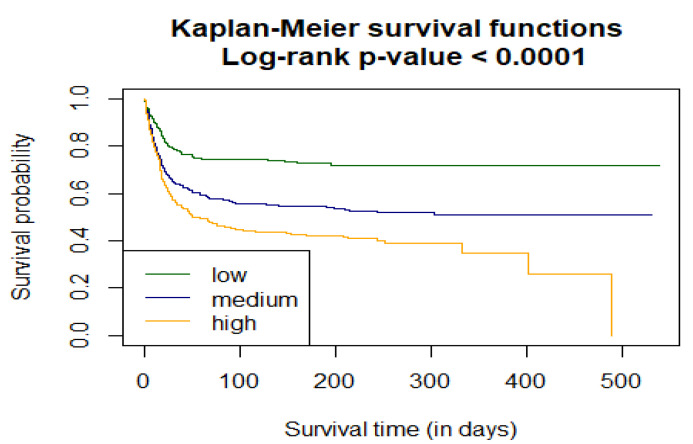
Analysis of the in-hospital probability of survival for the *low*, *medium*, and *high* C_2_HEST risk strata.

**Table 1 jcm-11-00992-t001:** Baseline characteristics of the study cohort after C_2_HEST risk stratification.

	Low Risk (0–1)	Medium Risk (2–3)	High Risk (>4)	OMNIBUS *p* Value	*p*-Value (for Post-Hoc Analysis)
Variables, Units (N)	Mean ± SD Min–Max (N) or n/N(% of Risk Category)	Mean ± SD Min–Max (N) or n/N (% of Risk Category)	Mean ± SD Min–Max (N) or n/N (% of Risk Category)		
**demographics**					
**Age, years** (1047)	69.0 ± 2.79 65–74 (376)	79.0 ± 8.11 65–100 (419)	80.3 ± 7.26 65–100 (252)	**<0.0001**	**<0.0001 ^a,b^** 0.082 **^c^**
**Male gender** (1047)	211/376 (56.11%)	172/419 (41.1%)	123/252 (48.8%)	**0.00012**	**<0.0001 ^a^** 0.2578 **^b^** 0.18001 **^c^**
**BMI, kg/m^2^** (207)	28.5 ± 4.59 20.05–40.4 (81)	28.57 ± 5.17 18.6–47.75 (66)	27.29 ± 5.39 16.41–45.82 (60)	0.30822	N/A
**Cigarette smoking** never/previous/current (1043)	348/376 (92.55%) 16/376 (4.26%) 12 376 (3.19%)	377/416 (90.63%) 25/416 (6.01%) 14/416 (3.37%)	203/251 (80.88%) 32/251 (12.75%) 16/251 (6.37%)	**<0.0001**	**1.0 ^a^** **0.00014 ^b^** **0.00412 ^c^**
**Co-morbidities**					
**Hypertension** (1047)	194/376 (51.6%)	296/419 (70.64%)	228/252 (90.48%)	**<0.0001**	**<0.0001 ^a,b,c^**
**DM** (1045)	106/376 (28.2%)	130/418 (29.7%)	103/251 (41.0%)	**0.00091**	1.0 **^a^** **0.0018 ^b^** **0.00578 ^c^**
**Dyslipidemia** (506)	104/152 (68.42%)	149/196 (76.02%)	133/158 (84.18%)	**0.0049**	0.4353 ^a^ **0.0052 ^b^** **0.02339 ^c^**
**AF/AFL** (1047)	32/376 (8.51%)	97/419 (23.15%)	124/252 (49.2%)	**<0.0001**	**<0.0001 ^a,b,c^**
**Previous coronary revascularization** (1047)	5/376 (1.33%)	26/419 (6.21%)	97/252 (38.5%)	**<0.0001**	**0.0023 ^a^** **<0.0001 ^b,c^**
**Previous MI** (1047)	8/376 (2.13%)	39/419 (9.31%)	103/252 (40.9%)	**<0.0001**	**<0.00011 ^a^** **<0.0001 ^b,c^**
**HF** (1047)	0/376 (0%)	32/419 (7.64%)	180/252 (71.43%)	**<0.0001**	**<0.0001^a,b,c^**
**Moderate or severe valvular heart disease or previous valve heart surgery** (1047)	6/376 (1.6%)	23/419 (5.49%)	48/252 (19.05%)	**<0.0001**	**0.0188 ^a^** **<0.0001 ^b,c^**
**PAD** (1047)	16/376 (4.26%)	25/419 (5.97%)	38/252 (15.05%)	**<0.0001**	1.0 **^a^** **<0.0001 ^b^** **0.00047 ^c^**
**Previous stroke/TIA** (1047)	25/376 (6.65%)	53/419 (12.65%)	52/252 (20.63%)	**<0.0001**	**0.0196 ^a^** **<0.0001 ^b^** **0.0243 ^c^**
**CKD** (1047)	25/376 (6.65%)	56/(13.37%)	82/252 (32.54%)	**<0.0001**	**0.00789 ^a^** **<0.0001 ^b,c^**
**Haemodialysis** (1047)	4/376 (1.06%)	13/492 (3.1%)	11/252 (4.37%)	**0.0332**	0.2464 **^a^** 0.0507 **^b^** 1.0 **^c^**
**Asthma** (1047)	12/376 (3.19%)	16/419 (3.82%)	10/252 (3.97%)	0.847	N/A
**COPD** (1047)	4/376 (1.06%)	20/419 (4.77%)	38/252 (15.08%)	**<0.0001**	**0.0134 ^a^** **<0.0001 ^b,c^**
**Thyroid disease**, none/hypothyroidism/ hyperthyroidism, (1047)	363/376 (96.5%) 12/376 (3.19%) 1/376 (0.27%)	368/419 (87.8%) 44/419 (10.5%) 7/419 (1.67%)	189/252 (75.0%) 58/252 (23.02%) 5/252 (1.98%)	**<0.0001**	**<0.0001 ^a,b^** **0.000018** ^c^

Continuous variables are presented as: mean ± SD, range (minimum–maximum) and number of non-missing values. Categorized variables are presented as: a number with a percentage. Information about the numbers with valid values is provided in the left column. Abbreviations: N—valid measurements, n—number of patients with parameter above cut-off point, SD—standard deviation, BMI—body mass index, DM—Diabetes mellitus, AF/AFL—Atrial fibrillation/flutter, MI—myocardial infarction, HF—Heart failure, PAD—Peripheral artery disease, TIA—transient ischemic attack, CKD—Chronic kidney disease, COPD—Chronic obstructive pulmonary disease, N/A—non-applicable, ^a^—*low-risk* vs. *medium-risk*, ^b^—*low-risk* vs. *high-risk*, ^c^—*medium-risk* vs. *high-risk*; Bold text—statistically significant values.

**Table 2 jcm-11-00992-t002:** Baseline characteristics of the study cohort-treatment applied before hospitalization.

	Low Risk (0–1)	Medium Risk (2–3)	High Risk (>4)	OMNIBUS *p*-Value	*p*-Value (for Post-Hoc Analysis)
Variables, Units (N)	n/N (% of Risk Category)	n/N (% of Risk Category)	n/N (% of Risk Category)		
**Treatment applied before hospitalization**					
**ACEI** (1047)	66/376 (17.55%)	96/419 (22.91%)	106/252 (42.06%)	**<0.0001**	0.2230 ^a^ **<0.0001 ^b,c^**
**ARBs** (1047)	31/376 (8.24%)	34/419 (8.11%)	29/252 (11.51%)	0.2721	N/A
**MRAs** (1047)	9/376 (2.39%)	26/419 (6.21%)	43/252 (17.06%)	**<0.0001**	**0.04377 ^a^** **<0.0001 ^b,c^**
**Sacubitril/valsartan** (1047)	1/376 (0.27%)	3/419 (0.72%)	1/252 (0.4%)	0.8502	N/A
**β-blocker** (1047)	93/376 (24.73%)	141/419 (33.65%)	143/252 (56.75%)	**<0.0001**	**0.02232 ^a^** **<0.0001 ^b,c^**
**Digitalis glycoside** (1047)	3/376 (0.8%)	5/419 (1.2%)	10/252 (3.97%)	**0.0129**	1.0 **^a^** **0.0259 ^b^** **0.0844 ^c^**
**Calcium channel blocker** **(non-dihydropiridines)** (1047)	6/376 (1.6%)	10/419 (2.39%)	13/252 (5.16%)	**0.0236**	1.0 ^a^ 0.0614 ^b^ 0.2718 ^c^
**Calcium channel blocker** **(dihydropiridines)** (1047)	44/376 (11.7%)	69/419 (16.47%)	69/252 (27.38%)	**<0.0001**	**<0.0001 ^a.b^** **0.00467 ^c^**
**α-adrenergic blocker** (1047)	45/376 (11.9%)	34/419 (8.11%)	39/252 (14.2%)	**<0.0001**	0.2065 ^a^ **<0.0001^b^** **0.0030 ^c^**
**Thiazide or thiazide-like diuretic** (1047)	30/376 (7.97%)	43/419 (10.26%)	32/252 (12.7%)	0.152	N/A
**Loop diuretic** (1047)	22/376 (5.85%)	50/419 (11.93%)	73/252 (28.97%)	**<0.0001**	**0.0127 ^a^** **<0.0001 ^b,c^**
**Statin** (1047)	59/376 (15.69%)	106/419 (25.3%)	113/252 (44.84%)	**<0.0001**	**0.0035 ^a^** **<0.0001 ^b,c^**
**Acetylsalicylic acid** (1047)	40/376 (10.64%)	79/419 (18.85%)	72/252 (28.57%)	**<0.0001**	**0.005 ^a^** **<0.0001 ^b^** **0.0143 ^c^**
**The second antiplatelet drug-P_2_Y_12_ inhibitor** (1047)	3/376 (0.8%)	6/419 (1.43%)	20/252 (7.94%)	**<0.0001**	1.0 ^a^ **<0.0001 ^b^** **0.00017 ^c^**
**LMWH** (1047)	30/376 (8.0%)	35/419 (8.35%)	24/252 (9.52%)	0.7856	N/A
**VKA** (1047)	4/376 (1.06%)	13/419 (3.1%)	21/252 (8.33%)	**<0.0001**	0.2464 ^a^ **<0.0001^b^** **0.00149 ^c^**
**NOAC** (1047)	9/376 (2.39%)	33/419 (7.88%)	49/252 (19.44%)	**<0.0001**	**0.003 ^a^** **<0.0001^b, c^**
**Insulin** (1047)	30/376 (7.98%)	23/419 (5.49%)	32/252 (12.7%)	**0.0041**	0.6203 **^a^** 0.2123 **^b^** **0.0049 ^c^**
**Metformin** (1047)	56/376 (14.89%)	58/419 (13.84%)	44/252 (17.46%)	0.4437	N/A
**SGLT2 inhibitor** (1047)	3/376 (0.8%)	5/419 (1.19%)	9/252 (3.57%)	**0.0274**	1.0 **^a^** 0.0504 ^b^ 0.1487 **^c^**
**Oral antidiabetics other than SGLT2 inhibitor and metformin** (1047)	19/376 (5.05%)	33/419 (7.88)	24/252 (9.52%)	0.0874	N/A
**Proton pump inhibitor** (1047)	30/376 (8.0%)	61/419 (14.56%)	80/252 (31.75%)	**<0.0001**	**0.0154 ^a^** **<0.0001 ^b,c^**
**Oral corticosteroid** (1047)	18/376 (4.79%)	21/419 (5.01%)	4/252 (1.59)	0.068	N/A
**Immunosuppression other than oral corticosteroid** (1047)	11/376 (2.93%)	17/419 (4.06%)	1/252 (0.37%)	**0.0194**	1.0 **^a^** 0.146 **^b^** **0.0284 ^c^**

Categorized variables are presented as: a number with a percentage. Information about the numbers with valid values is provided in the left column. Abbreviations: N—valid measurements. n—number of patients with parameter above the cut-off point. ACEI—angiotensin-converting-enzyme inhibitors. ARBs—angiotensin receptor blockers. MRAs—mineralocorticoid receptor antagonists. LMWH—low molecular weight heparin. VKA—vitamin K antagonists. NOAC—novel oral anticoagulants. SGLT2 inhibitors—sodium glucose co-transporter-2 inhibitors. N/A—non-applicable. ^a^—*low risk* vs. *medium risk*. ^b^—*low risk* vs. *high risk*. ^c^—*medium risk* vs. *high risk*. Bold text—statistically significant values.

**Table 3 jcm-11-00992-t003:** Patient-reported symptoms, vital signs, and abnormalities measured during physical examination at hospital admission in the studied cohort after C_2_HEST risk stratification.

	Low Risk (0–1)	Medium Risk (2–3)	High Risk (>4)	OMNIBUS *p*-Value	*p*-Value (for Post-Hoc Analysis)
Variables, Units (N)	Mean ± SD Min–Max (N) or n/N (% of Risk Category)	Mean ± SD Min–Max (N) or n/N (% of Risk Category)	Mean ± SD Min–Max (N) or n/N (% of Risk Category)		
**Patient-reported symptoms**					
**Cough** (1047)	94/376 (25%)	105/419 (25.06%)	64/252 (25.4%)	0.9931	N/A
**Dyspnoea** (1047)	153/376 (40.69%)	172/419 (41.05)	135/252 (53.57%)	**0.0019**	1.0 ^a^ **0.0059** **^b^** **0.0064 ^c^**
**Chest pain** (1047)	18/376 (4.79%)	29/419 (6.92%)	24/252 (9.52%)	0.068	N/A
**Hemoptysis** (1047)	1/376 (0.27%)	2/419 (0.48%)	4/252 (1.59%)	0.15	N/A
**Smell dysfunction** (1047)	11/376 (2.93%)	10/419 (2.29%)	4/252 (1.59%)	0.56	N/A
**Taste dysfunction** (1047)	9/376 (2.39%)	9/419 (2.15%)	6/252 (2.38%)	0.968	N/A
**Abdominal pain** (1047)	25/376 (6.65%)	23/419 (5.49%)	16/252 (6.35%)	0.78	N/A
**Diarrhoea** (1047)	29/376 (7.71%)	29/419 (6.92%)	17/252 (6.75%)	0.872	N/A
**Nausea and/or** **vomiting** (1047)	18/376 (4.79%)	23/419 (5.49%)	13/252 (5.16%)	0.905	N/A
**Measured vital signs**					
**Body temperature** °C (522)	36.98 ± 0.87 35.0–40.0 (189)	36.89 ± 0.9 35.0–40.0 (203)	36.94 ± 0.89 35.2–40.0 (130)	0.572	N/A
**Heart rate** beats/minute (823)	86.64 ± 16.72 60–150 (280)	84.06 ± 16.52 50–160 (325)	84.75 ± 18.92 36–170 (218)	0.156	N/A
**Respiratory rate** breaths/minute (152)	18.25 ± 6.1 12–50 (52)	18.79 ± 5.71 12–45 (58)	19.52 ± 6.33 12–50 (42)	0.619	N/A
**SBP** mmHg (832)	134.92 ± 23.13 60–237 (283)	134.55 ± 25.87 50–270 (327)	134.0 ± 24.39 70–210 (222)	0.912	N/A
**DBP** mmHg (826)	78.23 ± 13.8 40–150 (282)	77.54 ± 13.68 40–157 (322)	75.54 ± 15.43 40–143 (222)	0.1197	N/A
**SpO2 on room air, %** (FiO_2_ = 21%) (587)	90.5 ± 7.85 50–100 (194)	89.2 ± 9.74 50–100 (238)	90.02 ± 8.48 50–99 (155)	0.3383	N/A
**Abnormalities detected during physical examination**					
**Cracles** (1047)	56/376 (14.89%)	84/419 (20.05%)	58/252 (23.02%)	**0.029**	0.21 ^a^ **0.0391 ^b^** 1.0 ^c^
**Wheezing** (1047)	35/376 (9.31%)	51/419 (12.17%)	61/252 (24.21%)	**<0.0001**	**0.071 ^a^** **<0.0001** ** ^b^ ** **0.00024 ^c^**
**Pulmonary congestion** (1047)	66/376 (17.55%)	90/419 (21.48%)	71/252 (28.17%)	**0.0066**	**0.5784 ^a^** 0.066 **^b^** 0.1831 ^c^
**Peripheral oedema** (1047)	27/376 (7.18%)	48/419 (11.46%)	47/274 (18.65%)	**<0.0001**	**0.1581 ^a^** **<0.0001 ^b^** **0.04 ^c^**
**VES-13, points**				0.067	N/A
mean ± SD	4.24 ± 2.99	5.58 ± 3.3	6.54 ± 2.89		
min–max	1–9	1–12	3–13		
N = 75	17	36	22		
**GCS, points**				0.305	N/A
mean ± SD	14.57 ± 1.75	14.38 ± 1.81	14.18 ± 2.27		
min–max	3–15	3–15	3–15		
N = 402	133	160	109		

Continuous variables are presented as: mean ± SD, range (minimum–maximum) and number of non-missing values. Categorized variables are presented as: a number with a percentage. Information about the numbers with valid values is provided in the left column. Abbreviations: SD—standard deviation, OMNIBUS—analysis of variance, N—valid measurements, n—number of patients with parameter above cut-off point, SBP—Systolic blood pressure, DBP—Diastolic blood pressure; VES—Vulnerable Elders Survey, GCS—Glasgow Coma Scale, *^a^—low risk vs. medium risk, ^b^—low risk vs. high risk, ^c^—medium risk vs. high risk*.

**Table 4 jcm-11-00992-t004:** Laboratory parameters measured during the hospitalization in the studied cohort.

Parameter	Time of Assessment	Units	Low Risk (0–1)	Medium Risk (2–3)	High Risk (>4)	OMNIBUS *p*-Value	*p*-Value for Post-Hoc Analysis
			Mean ± SD Min–Max (N) or n/N (% of Risk Category) (N)	Mean ± SD Min–Max (N) or n/N (% of Risk Category) (N)	Mean ± SD Min–Max (N) or n/N (% of Risk Category) (N)		
**Complete Blood Count (CBC)**							
Leucocytes (1020)	On admission	10^3^/µL	8.8 ± 8.75 0.51–150.93 (364)	9.55 ± 12.26 0.51–215.97 (410)	9.37 ± 8.13 1.19–99.73 (246)	0.5472	N/A
(1020)	On discharge		9.17 ± 5.97 0.67–53.2 (364)	10.83 ± 17.42 0.44–314.44 (410)	10.2 ± 7.38 1.19–58.49 (246)	0.063	N/A
**Lymphocytes** (697)	On admission	10^3^/µL	1.17 ± 1.65 0.06–24.82 (237)	1.16 ± 1.13 0.11–12.1 (278)	1.44 ± 5.78 0.09–78.58 (182)	0.8223	N/A
(677)	On discharge		1.57 ± 1.02 0.06–9.03 (237)	1.48 ± 1.97 0.05–26.71 (278)	1.55 ± 5.04 0.14–66.97 (182)	0.787	N/A
**Haemoglobin** (1020)	On admission	g/dL	13.11 ± 2.12 3.9–18.3 (364)	12.55 ± 2.33 4.5–18.9 (410)	11.93 ± 2.49 5.3–18.8 (246)	**<0.0001**	**0.001^a^** **<0.0001^b^** **0.005 ^c^**
(1020)	On discharge		12.5 ± 2.18 7.1–18.3 (364)	11.91 ± 2.33 4.5–18.9 (410)	11.56 ± 2.35 5.5–17.6 (246)	**<0.0001**	**0.0008 ^a^** **<0.0001^b^** 0.154 **^c^**
**Platelets** (1020)	On admission	10^3^/µL	245.79 ± 110.26 0–671 (364)	228.85 ± 114.82 3–740 (410)	216.78 ± 94.0 8–578 (246)	**0.0023**	0.092 **^a^** **0.002 ^b^** 0.31 **^c^**
(1020)	On discharge		272.04 ± 119.9 6–720 (364)	241.95 ± 118.27 3–694 (410)	211.63 ± 97.47 4–592 (246)	**<0.0001**	**<0.001 ^a^ <0.0001^b^ 0.001 ^c^**
**Acid-base balance in the arterial blood gas**							
**PH** (175)	On admission		7.43 ± 0.08 7.2–7.54 (43)	7.43 ± 0.07 7.1–7.54 (74)	7.41 ± 0.08 7.09–7.54 (58)	0.3236	N/A
**PaO2** (175)	On admission	<60 mmHg respiratory insufficiency	27/43 (62.79%)	44/74 (59.46%)	34/58 (58.62%)	0.9073	N/A
		≥60 mmHg	16/43 (37.21%)	30/74 (40.54%)	24/58 (41.38%)		
			76.3 ± 34.37 26.8–100 (43)	75.46 ± 48.27 28.6–100 (74)	72.91 ± 36.32 23.7–100 (58)	0.8821	N/A
**PaCO2** (175)	On admission	≥45 mmHg hypercapnia	7/43 (16.28%)	8/74 (10.81%)	10/58 (17.24%)	0.5265	N/A
		<45 mmHg	36/43 (83.72%)	66/74 (89.19%)	48/58 (82.76%)		
			36.57 ± 8.0 25.2–61.4 (43)	36.58 ± 8.02 23–67 (74)	38.9 ± 11.43 19.7–88.4 (58)	0.3899	N/A
**HCO3 standard** (171)	On admission	mmol/L	25.05 ± 3.7 12.1–30.7 (42)	24.47 ± 4.17 14.3–39.5 (71)	24.43 ± 4.72 13.5–38.6 (58)	0.6908	N/A
**BE** (74)	On admission		1.64 ± 3.08 (–)3.3–7.1 (17)	2.15 ± 4.88 (–)12.5–15.7 (37)	2.41 ± 5.55 (–)7.4–14.6 (20)	0.8345	N/A
**Lactates** (157)	On admission		2.7 ± 2.28 0.7–12.8 (38)	2.03 ± 0.85 0.5–5.7 (66)	2.55 ± 1.91 0.6–12.0 (53)	0.0602	N/A
**Electrolytes. inflammatory and iron biomarkers**							
**Na** (1015)	On admission	mmol/L	137.89 ± 5.16 106–159 (362)	137.81 ± 7.37 101–175 (407)	138.1 ± 6.98 108–174 (246)	0.8784	N/A
**K** (1018)	On admission	mmol/L	4.07 ± 0.66 2.0–7.5 (363)	4.12 ± 0.7 2.4–6.08 (409)	4.27 ± 0.8 2.53–8.7 (246)	**0.0066**	0.602 **^a^** **0.005 ^b^** 0.044 **^c^**
**CRP** (1015)	On admission	mg/L	93.03 ± 91.05 0.32–496.98 (361)	84.51 ± 88.21 0.29–538.55 (408)	76.19 ± 80.82 0.4–390.94 (246)	0.0574	N/A
**Procalcitonin** (748)	On admission	ng/mL	1.36 ± 6.32 0.01–61.28 (266)	2.02 ± 13.06 0.01–196.04 (289)	1.486.25 0.01–60.77 (193)	0.7464	N/A
**IL-6** (330)	On admission	pg/mL	66.81 ± 155.27 2–1000 (141)	41.58 ± 53.49 2–398 (120)	62.78 ± 98.77 2–421 (69)	0.0751	N/A
**D-dimer** (804)	On admission	µg/L	4.56 ± 13.34 0.18–118.32 (298)	6.37 ± 16.17 0.2–127.24 (319)	5.77 ± 17.97 0.22–128.0 (187)	0.301	N/A
**Prothrombin rate** (958)	On admission	%	82.6 ± 15.73 37–128 (343)	79.43 ± 21.33 7–131 (382)	70.49 ± 26.47 2–124 (252)	**<0.0001**	**0.058 ^a^** **<0.0001 ^b,c^**
**INR** (958)	On admission	>1.5	12/344 (3.49%)	40/381 (10.5%)	55/233 (23.61%)	**<0.0001**	**0.0014 ^a^** **<0.0001 ^b,c^**
**aPTT** (927)	On admission	>60 s	3/331 (2.11%)	6/369 (1.63%)	10/227 (4.41%)	0.092	N/A
**Urea** (970)	On admission	mg/dL	57.13 ± 46.17 8–307 (345)	67.31 ± 49.77 12–353 (389)	77.66 ± 52.55 12–369 (236)	**<0.0001**	**0.012 ^a^** **<0.0001^b^** **0.04 ^c^**
**Creatinine** (1017)	On admission	mg/dL	1.3 ± 1.31 0.49–14.77 (361)	1.42 ± 1.15 0.48–9.56 (410)	1.75 ± 1.54 0.44–11.3 (246)	**0.0009**	**0.349 ^a^** **0.0006 ^b^** **0.012 ^c^**
(1017)	On discharge		1.16 ± 1.04 0.44–14.82 (361)	1.39 ± 1.2 0.43–9.09 (410)	1.59 ± 1.34 0.43–9.27 (246)	**<0.0001**	**0.01 ^a^** **<0.0001^b^** 0.134 **^c^**
**eGFR** (1017)	On admission	ml/min/1.73 m^2^	71.33 ± 27.92 3–170 (361)	6.29 ± 27.45 4–137 (410)	52.99 ± 28.95 5–180 (246)	**<0.0001**	**<0.0001 ^a,b^** **0.004 ^c^**
**Total protein** (334)	On admission	g/L	5.99 ± 0.8 3.8–7.7 (100)	5.87 ± 0.89 3.6–8.2 (123)	5.73 ± 0.9 3.3–8.2 (111)	0.0909	N/A
**Albumin** (363)	On admission	g/L	3.16 ± 0.54 1.7–4.4 (116)	3.09 ± 0.55 1.1–4.4 (130)	2.95 ± 0.62 0.7–4.9 (117)	**0.0191**	0.528 **^a^** **0.014 ^b^** 0.151 **^c^**
**AST** (740)	On admission	IU/L	70.12 ± 177.91 5–2405 (257)	69.44 ± 281.44 7–4776 (290)	90.01 ± 339.29 8–3866 (193)	0.7435	N/A
**ALT** (821)	On admission	IU/L	55.67 ± 113.23 4–1411 (285)	49.33 ± 206.01 4–3700 (329)	54.0 ± 149.9 5–1361 (207)	0.8911	N/A
**Bilirubin** (736)	On admission	U/L	0.91 ± 1.34 0.3–15.1 (257)	0.83 ± 0.74 0.2–9.2 (296)	0.88 ± 0.72 0.1–6.6 (183)	0.6838	N/A
**LDH** (623)	On admission	U/L	466.34 ± 561.39 129–7100 (232)	389.48 ± 191.8 44–1172 (237)	453.63 ± 768.4 71–9505 (154)	0.0978	N/A
**Cardiac biomarkers**							
**BNP** (244)	On admission	pg/mL	198.97 ± 295.09 1.7–1674 (71)	411.54 ± 765.61 3–4890.6 (85)	950.94 ± 2052.17 12.4–13,368.4 (88)	**0.00051**	0.052 **^a^** **0.003 ^b^** 0.059 **^c^**
(244)	On discharge		187.85 ± 236.76 1.7–1130.8 (71)	456.81 ± 1251.89 3–10,662.8 (85)	894.93 ± 1965.08 11.9–13,368.4 (88)	**0.00104**	0.133 **^a^** **0.003 ^b^** 0.188 **^c^**
**NT-proBNP** (239)	On admission	ng/mL	2647.61 ± 91,184.03 12–70,000 (63)	8356.29 ± 14,376.9 49.6–70,000 (87)	13,371.9 ± 18,707.7 119.6–70,000 (89)	**<0.0001**	**0.01^a^** **<0.0001 ^b^** **0.116 ^c^**
(239)	On discharge		2591.46 ± 6818.7 12–35,000 (63)	9044.29 ± 15,277.1 49.6–70,000 (87)	12,370.9 ± 16,896.4 119.6–70,000 (89)	**<0.0001**	**0.002 ^a^** **<0.0001 ^b^** 0.359 **^c^**
**Troponin T** *normal value*: *F* ≤ *15.6* pg/mL *M* ≤ *34.2* pg/mL (665)	On admission	pg/mL	171.38 ± 899.58 0.2–11,398.7 (228)	1968.15 ± 12,515.9 2.0–125,593 (263)	658.56 ± 2437.77 3.3–21,022.9 (174)	**0.0037**	0.055 **^a^** **0.034 ^b^** 0.226 **^c^**
**Troponin T** (665)	On discharge		152.13 ± 890.6 0.2–12,391.6 (228)	1490.76 ± 9509.94 1.5–109,360 (263)	664.38 ± 2887.8 1.8–29,828.3 (174)	**0.0074**	0.062 **^a^** 0.064 ^b^ 0.385 **^c^**
**LDL-cholesterol**. (268)	On admission	mg/dL	87.7 ± 40.22 6–205 (80)	89.79 ± 41.8 23–230 (106)	75.59 ± 42.83 14–210 (82)	0.0554	N/A
**Hormones**							
**TSH** (474)	On admission	mIU/L	1.35 ± 1.52 0.07–14.08 (149)	1.35 ± 1.69 0.01–12.1 (188)	2.24 ± 4.09 0–38.24 (137)	**0.049**	1.0 **^a^** **0.045 ^b^** 0.046 **^c^**
**fT4 n** (194)	On admission	pmol/L	12.78 ± 2.27 6.68–19.05 (58)	13.03 ± 3.4 7.56–36.6 (79)	13.48 ± 4.17 7.87–35.46 (57)	0.5257	N/A
**fT3** (176)	On admission	pmol/L	2.08 ± 0.63 1.2–4.01 (57)	1.88 ± 0.77 0.95–4.45 (71)	1.93 ± 0.97 0.95–6.85 (48)	0.2684	N/A

Continuous variables are presented as: mean ± SD. range (minimum–maximum) and number of non-missing values. Categorized variables are presented as: a number with a percentage. Information about the numbers with valid values is provided in the left column. Abbreviations: N—valid measurements. n—number of patients with parameter above cut-off point. SD—standard deviation. N/A—non-applicable. *^a^—low risk vs. medium risk. ^b^—low risk vs. high risk. ^c^—medium risk vs. high risk*.

**Table 5 jcm-11-00992-t005:** Therapies applied during the hospitalization in the studied cohort.

Variables. Units (N)	Low Risk (0–1)	Medium Risk (2–3)	High Risk (>4)	OMNIBUS *p* Value	*p* Value (for Post-Hoc Analysis)
	n/N (% of Risk Category)	n/N (% of Risk Category)	n/N (% of Risk Category)		
**Applied treatment and procedures**					
**Systemic corticosteroid** (1047)	212/376 (56.38%)	211/419 (50.36%)	129/252 (51.19%)	0.2021	N/A
**Convalescent plasma** (1047)	56/376 (14.89%)	32/419 (7.64%)	27/252 (10.71%)	**0.0048**	**0.005 ^a^** 0.48885 **^b^** 0.6648 **^c^**
**Tocilizumab** (1047)	6/376 (1.6%)	2/419 (0.48%)	1/252 (0.4%)	0.2223	N/A
**Remdesivir** (1047)	68/376 (18.09%)	59/419 (14.08%)	32/252 (12.7%)	0.1312	N/A
**Antibiotic** (1047)	230/376 (61.17%)	264/419 (63.01%)	175/252 (69.44%)	0.09451	N/A

Categorized variables are presented as: a number with a percentage. Information about the numbers with valid values is provided in the left column. Abbreviations: N—valid measurements. n—number of patients with parameter above cut-off point. SD—standard deviation. N/A—non-applicable. *^a^—low risk vs. medium risk. ^b^—low risk vs. high risk. ^c^—medium risk vs. high risk*; Bold text—statistically significant values Bold text-statistically significant values.

**Table 6 jcm-11-00992-t006:** Applied treatment and procedures.

	Low Risk (0–1)	Medium Risk (2–3)	High Risk (>4)	OMNIBUS *p*-Value	*p*-Value (for Post-Hoc Analysis)
Variables, Units (N)	Mean ± SD Min–Max (N) or n/N (% of Risk Category)	Mean ± SD Min–Max (N) or n/N (% of Risk Category)	Mean ± SD Min–Max (N) or n/N (% of Risk Category)		
**Applied treatment and procedures**					
**The most advanced respiratory support applied during the hospitalization** (1047)					
no oxygen high flow nasal cannula (non-invasive ventilation) invasive ventilation	159/376 (42.29%) 21/376 (5.59%) 47/376 (12.5%)	168/418 (40.19%) 36/418 (8.61%) 41/418 (9.81%)	86/252(34.13%) 22/252 (8.73%) 19/252 (7.54%)	**0.0415**	0.9925 ^a^ **0.0188 ^b^** 0.6137 **^c^**
**Oxygenation parameters from the period of qualification for advanced respiratory support:**					
SpO2 (284) Respiratory rate, breaths/minute (62)	87.35 ± 9.89 50–99 (86) 25.64 ± 6.96 14–40 (14)	86.19 ± 9.79 55–99 (116) 30.11 ± 14.0 13–66 (27)	85.53 ± 9.86 59–99 (82) 29.52 ± 13.19 14–72 (21)	0.4815 0.3147	N/A
**Duration of mechanical ventilation, days** (616)	1.89 ± 5.52 0–39 (222)	1.4 ± 5.18 0–51 (240)	1.14 ± 4.07 0–29 (154)	0.3134	N/A
**Therapy with catecholamines** (1047)	44/376 (11.7%)	36/419 (8.6%)	37/252 (14.7%)	**0.0486**	0.5433 ^a^ 0.9949 ^b^ 0.0601 ^c^
**Coronary angiography** (1047)	5/376 (1.3%)	10/419 (2.4%)	7/252 (2.8%)	0.4036	N/A
**Coronary revascularization** (1047)	4/376 (1.1%)	9/419 (2.1%)	6/252 (2.4%)	0.3893	N/A
**Hemodialysis** (1047)	16/376 (4.3%)	11/419 (2.6%)	11/252 (4.7%)	0.3644	N/A

Continuous variables are presented as: mean ± SD, range (minimum–maximum) and number of non-missing values. Categorized variables are presented as: a number with a percentage. Information about the numbers with valid values is provided in the left column. Abbreviations: N—valid measurements, n—number of patients with parameter above cut-off point, SD—standard deviation, ANOVA—analysis of variance, N/A—non-applicable, *^a^—low risk vs. medium risk, ^b^—low risk vs. high risk, ^c^—medium risk vs. high risk*.

**Table 7 jcm-11-00992-t007:** Total and in-hospital all-cause mortality in the C_2_HEST risk strata.

	Low Risk (0–1)	Medium Risk (2–3)	High Risk (>4)		
Variables, Units (N)	Mean ± SD Min–Max (N) or n/N (% of Risk Category)	Mean ± SD Min–Max (N) or n/N (% of Risk Category)	Mean ± SD Min–Max (N) or n/N (% of Risk Category)	OMNIBUS *p*-Value	*p*-Value (for Post-Hoc Analysis)
**All-cause mortality rate**					
**In-hospital** **mortality** (1047)	54/376 (14.4%)	101/419 (24.1%)	90/252 (35.7%)	**<0.0001**	**0.00223 ^a^** **<0.0001 ^b^** **0.005 ^c^**
**3-month** **mortality** (1047)	97/376 (25.8%)	182/419 (43.4%)	137/252 (54.4%)	**<0.0001**	**<0.0001 ^a, b^** **0.023 ^c^**
**6-month** **mortality** (810)	102/260 (39.2%)	190/330 (57.6%)	145/220 (65.9%)	**<0.0001**	**<0.0001 ^a, b^** 0.1832 **^c^**

Categorized variables are presented as: a number with a percentage. Abbreviations: N—valid measurements, n—number of patients with parameter above cut-off point, SD—standard deviation, ANOVA—analysis of variance, N/A—non-applicable, *^a^—low risk vs. medium risk, ^b^—low risk vs. high risk, ^c^—medium risk vs. high risk*. Bold text—statistically significant values.

**Table 8 jcm-11-00992-t008:** The total all-cause-death Hazard Ratios for C_2_HEST risk stratification.

Total Death			
Overall	HR	95%CI	*p*-Value
	1.21	NA	NA
**Risk strata**			
**Low risk vs.** **Medium risk**	1.94	1.531–2.467	**<0.0001**
**Low risk vs.** **High risk**	2.70	2.104–3.473	**<0.0001**

**Table 9 jcm-11-00992-t009:** Associations of individual C_2_HEST score components with mortality.

	Component	HR	CI min.	CI max.	*p*-Value
All-cause mortality	Coronary artery disease	1.457	1.143	1.856	0.0023
	COPD	1.128	0.787	1.615	0.5118
	Age > 75	1.852	1.528	2.243	< 0.0001
	Thyroid disease	0.781	0.579	1.052	0.1041
	Hypertension	0.867	0.706	1.065	0.1738
	HFrEF	1.412	1.117	1.783	0.0038

COPD—Chronic obstructive pulmonary disease, HFrEF—heart failure with reduce ejection fraction.

**Table 10 jcm-11-00992-t010:** The Log-rank statistics for matching the C2HEST risk strata for in-hospital mortality.

	h2	h3	h4	h5	h6	h7	h8
m1		54.9289	45.309	36.9829	22.3874	19.5331	4.391
m2		55.5515	**64.8647**	62.9116	55.5126	54.8399	7.4052
m3			43.3222	40.8103	33.8943	33.4495	5.7835
m4				36.9734	36.2196	36.2402	5.9926
m5					25.3749	24.3364	4.862
m6						5.0713	2.2214
m7							0.7235

m—medium. h—high. Bold text—highest statistical significant

**Table 11 jcm-11-00992-t011:** Clinical non-fatal events and hospitalization outcomes in the C_2_HEST risk strata.

	Low Risk (0–1)	Medium Risk (2–3)	High Risk (>4)	OasMNIBUS *p*-Value	*p*-Value (for Post-Hoc Analysis)
Variables, Units (N)	Mean ± SD Min–Max (N) or n/N (% of Risk Category)	Mean ± SD Min–Max (N) or n/N (% of Risk Category)	Mean ± SD Min–Max (N) or n/N (% of Risk Category)		
**Hospitalization**					
**Duration of hospitalization,****days** (1047)	13.45 ± 115.35 1–131 (376)	13.13 ± 13.98 1–124 (419)	15.79 ± 15.77 1–121 (252)	0.07693	N/A
**Admission at ICU** (1047)	46/376 (12.2%)	32/419 (7.6%)	24/252 (9.5%)	0.0916	N/A
**End of hospitalization** (1047)					
death discharge home–full recovery transfer to another hospital–worsening transfer to another hospital–in recovery	54/376 (14.4%) 210/376 (55.9%) 57/376 (15.2%) 55/376 (14.6%)	101/419 (24.1%) 176/419 (42.0%) 87/419 (20.8%) 55/419 (13.1%)	90/252 (36.6%) 92/252 (36.6%) 44/252 (17.5%) 26/252 (10.3%)	**<0.0001**	**0.000283 ^a^** **<0.0001 ^b^** **0.04329 ^c^**
**Clinical events**					
**Aborted cardiac arrest** (1047)	9/376 (2.4%)	1/419 (0.2%)	5/252 (2.0%)	**0.0127**	**0.0242 ^a^** 1.0 **^b^** 0.0906 **^c^**
**Shock** (1047) hypovolemic shock cardiogenic shock septic shock	39/376 (10.4%) 9/376 (2.4%) 0/376 (0%) 31/376 (8.2%)	37/419 (8.8%) 6/419 (1.4%) 9/419 (2.2%) 25/419 (6.0%)	29/252 (11.6%) 5/252 (2.0%) 13/252 (5.2%) 19/252 (7.5%)	0.515 0.6274 **<0.0001** 0.4454	N/A N/A **0.0349 ^a^** **<0.0001 ^b^** 0.1734 **^c^** N/A
**Venous thromboembolic disease** (1047)	28/376 (7.5%)	28/419 (6.7%)	15/252 (0.8%)	0.7619	N/A
**Pulmonary embolism** (1047)	24/376 (6.4%)	25/419 (6.0%)	13/252 (6.0%)	0.972	N/A
**Deep vein thrombosis** (1047)	1/376 (0.3%)	1/419 (0.2%)	0/252 (0.0%)		
**MI** (1047)	4/376 (1.1%)	9/419 (2.2%)	7/252 (2.8%)	0.2629	N/A
**Acute HF** (1047)	3/376 (0.8%)	14/419 (3.3%)	44/252 (17.5)	**<0.0001**	0.0773 **^a^** **<0.0001 ^b,c^**
**Stroke/TIA** (1047)	6/376 (1.6%)	16/419 (3.8%)	7/252 (2.8%)	0.1623	N/A
**Pneumonia** (1047)	224/376 (59.6%)	264/419 (63.0%)	168/252 (66.7%)	0.1939	N/A
**SIRS** (1040)	37/373 (9.9%)	38/416 (9.1%)	33/251 (13.1%)	0.2412	N/A
**Sepsis** (405)	2/137 (1.5%)	7/153 (4.6%)	6/115 (5.2%)	0.1866	N/A
**Acute kidney injury** (1047)	37/376 (9.8%)	59/419 (14.1%)	53/252 (21.0%)	**0.000432**	0.2546 ^a^ **0.000422 ^b^** 0.0769 **^c^**
**Acute liver dysfunction** (981)	7/352 (2.0%)	17/398 (4.3%)	13/231 (5.6%)	0.0623	N/A
**MODS** (1047)	6/376 (1.6%)	5/419 (1.2%)	6/252 (2.4%)	0.4735	N/A
**LA** (157)	5/38 (13.2%)	5/66 (7.6%)	6/53 (11.3%)	0.6287	N/A
**Hyperlactaemia** (157)	28/38 (73.7%)	43/66 (65.2%)	32/53 (60.4%)	0.4174	N/A
**Bleedings** (1047)	19/376 (5.2%)	21/419 (5.2%)	22/252 (8.7%)	0.09539	N/A
**Intracranial bleeding** (1047)	2/376 (0.5%)	8/419 (1.9%)	1/252 (0.4%)	0.1166	N/A
**Respiratory tract bleeding** (1047)	6/376 (1.6%)	2/419 (0.5%)	6/252 (2.4%)	0.0833	N/A
**Gastrointestinal tract bleeding** (1047)	9/376 (2.4%)	8/419 (1.9%)	10/252 (4.0%)	0.2667	
**Urinary tract bleeding** (1047)	3/1418 (0.8%)	4/492 (1.0%)	5/252 (2.0%)	0.4017	N/A

Continuous variables are presented as: mean ± SD range (minimum–maximum) and number of non-missing values. Categorized variables are presented as: a number with a percentage. Abbreviations: N—valid measurements, n—number of patients with parameter above cut-off point, SD—standard deviation, ANOVA—analysis of variance, ICU—intensive care unit, MI—myocardial infarction, HF—heart failure, TIA-transient ischemic attack, SIRS—systemic inflammatory response syndrome, MODS—multiple organ dysfunction syndrome, LA—lactic acidosis, N/A—non-applicable, *^a^—low risk vs. medium risk, ^b^—low risk vs. high risk, ^c^—medium risk vs. high risk*.

**Table 12 jcm-11-00992-t012:** Discriminatory performance of the C_2_HEST score on the clinical events.

CLINICAL EVENT	AUC	SENSITIVITY	SPECIFICITY
**End of hospitalization–full recovery**	**0.584**	**0.439**	**0.708**
**End of hospitalization–deterioration**	**0.511**	**0.697**	**0.371**
**End of hospitalization–rehabilitation**	**0.459**	**0.015**	**0.986**
**End of hospitalization–death**	**0.633**	**0.616**	**0.580**
**All-cause shock**	0.521	0.914	0.156
**Hypovolemic shock**	0.477	0.900	0.149
**Cardiogenic shock**	**0.774**	**1.000**	**0.367**
**Septic shock**	0.498	0.920	0.154
**Pulmonary embolism**	0.491	0.087	0.941
**Deep vein thrombosis**	0.463	0.222	0.940
**Venous thromboembolic disease**	0.487	0.085	0.941
**Myocardial infarction**	0.609	0.650	0.537
**Myocardial injury**	0.612	0.769	0.396
**Acute heart failure**	**0.824**	**0.721**	**0.789**
**Stroke/TIA**	0.582	0.655	0.539
**SIRS**	0.531	0.537	0.539
**Sepsis**	0.655	0.800	0.495
**Acute kidney injury**	**0.605**	**0.624**	**0.560**
**Acute liver dysfunction**	0.626	0.811	0.365
**MODS**	0.577	0.647	0.537
**All bleedings**	0.579	0.354	0.766
**Intracranial bleeding**	0.581	0.727	0.537
**Respiratory tract bleeding**	0.556	0.428	0.762
**Upper-GI-tract bleeding**	0.583	0.300	0.873
**Lower-GI-tract-bleeding**	0.529	0.571	0.642
**Urinary tract bleeding**	0.619	0.416	0.873
**Pneumonia**	0.543	0.489	0.5729

TIA—transient ischemic attack, SIRS—systemic inflammatory response syndrome MODS—multiple organ dysfunction syndrome, GI—gastrointestinal. Bold text—statistically significant values.

## Data Availability

The datasets used and/or analyzed during the current study are available from the corresponding author upon reasonable request.

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
