# Peer review of "Mortality Predictive Value of the C2HEST Score in Elderly Subjects with COVID-19—A Subanalysis of the COLOS Study"

_jcm, 2022, doi:10.3390/jcm11040992_

Round 1
Reviewer 1 Report
Mortality predictive value of the C2HEST score in elderly subjects with COVID-19- a subanalysis of the COLOS study. Reviewer comments.
In this study, Rola et al use a subset of elderly population among the COLOS registry of hospitalized COVID-19 patients to evaluate CHEST score ability to predict mortality among these patients. The main result is the CHEST score can delineate subgroups of elderly COVID-19 patients with significantly different in-hospital and 3&6-month mortality. This result is supported by multiple statistical analysis, which is one of the strengths of this study. However, there are major drawbacks which should be addresses.
Major comments:
- Introduction- The introduction is well written but misleading. The authors explain nicely the rational of evaluating CHEST score in COVID-19 patients, based on their insight that SCORE components are identical to the main risk factors predicting mortality in these patients. However, two of the main risk factors which the authors themselves mention are DM and obesity which are not included in CHEST score. Thus, I would think that adding these 2 main risk factors would further improve ability to delineate subgroups with different mortality rates. Notably, both DM and BMI are not well differentiated by post-hoc analysis of CHEST score groups. Accordingly, I think as the authors themselves suggest, that a clinical score of all risk factors would be best and I urge the authors to test a new score including both BMI (with some cutoff value to enable category) and DM. Indeed, the preset CHEST score delineated in-hospital mortality between 14% in the low risk and 38% in the high-risk but thinking further this in not always helpful in decisions regarding treatment as 14% mortality is not negligible. I can foresee that a score including DM and BMI might further delineate mortality to < 10% in low risk and >50% in the high risk.
- Methods- the authors say that age >65 was the main inclusion criteria. As I understand from the work it was the sole criteria. If indeed, should say explicitly. If I misunderstand, other inclusion criteria should be specified.
- Results- There are few main issues which are major drawback to the current study.
a) There is no consideration of the initial COVID severity on admission. Notably there were significantly more patients with dyspnea and wheezing among the CHEST high risk group. I would think that the major reason for this in not necessarily their baseline cardiac status but rather their COVID respiratory involvement. Multiple studies have proven that this is one of the most critical determinants of both the need for ICU admission and overall prognosis. Thus, I do not except the author writing in the discussion that the lack of significant difference of inflammatory markers level between CHEST groups implies that all patients admitted were of same COVID severity.
b) With the same notion as above- the analysis of respiratory support needed during hospitalization is inaccurate and misleading as not all patients are included in this analysis (table 6). Notably, this should include all patients as it implies to all levels of respiratory status. Indeed, re-analysis including all patients will help understand the current pardox of more invasive ventilation in the CHEST low risk group. This will also clarify patients' corona respiratory severity (which as I commented previously, should be included in a multivariate analysis for mortality).
c) ICU admission is relatively neglected in the paper and appears only regarding secondary analysis. However, need for ICU is the most emphasized issue in most COVID papers! The fact that the high risk CHEST score group had the lowest percent of ICU admission is worrisome. One can not rule out the possibility that the HIGH CHEST score group was the group with more sever COVID illness but nevertheless the group with least invasive monitoring, support and attention (within ICU). If so, their increased mortality could be attributed to this issue rather to CHEST score prediction. To try and adjust for this issue, I think the authors need to re-analyze death in context of yes/no ICU admission. Moreover, the authors should dedicate a paragraph in discussion to explain possible reasons for the relatively low percent of overall patients admitted to ICU compared with prior major studies reporting up to 25% of ICU admission among hospitalized COVID-19 patients.
d) The incidence of mortality according to CHEST risk groups is well presented as well as the KM analysis. However, in the multivariate one, the authors do not write what parameters were included in this analysis. Here I suggest that both ICU admission as well as COVID initial severity should be included as well.
On the whole, I think this is a novel and important work. However I urge the authors to evaluate DM and obesity within the scoring system and to adjust analysis to consider initial COVID severity and IC admission as well.
Author Response
We would like to thank the Reviewer for an in-depth analysis of the manuscript and for pivotal comments provided, which have resulted in a significant improvement of this manuscript.
Ad 1.We would like to thank the Reviewer for an important suggestion regarding the role of DM and obesity in predicting outcome among patients with COVID-19. We fully agree with the Reviewer that these risk-factors are important and might improve ability to delineate subgroups with different mortality rates.However, the basis of our study was the assumption of the use of a simple, well-validated, and particularly well-known from an everyday clinical score, which would allow for prompt use in everyday clinical practice. Aware of the special prognostic role of obesity diabetes in a patient with Covid-19, we have performed an additional sub-analysis of the COLOS study in conjunction with the C2HEST score, which is now presented in the supplementary file (the C2HEST-OD score).
We have also commented on this issue in the Discussion section, as follows:
Since obesity and diabetes consititue important comorbidities which could affect the COVID-19 outcome, including these parameters to this analysis could further improve the prognostic value of such modified C2HEST-OD which was presented in the supplementary material. Nevertheless, as specified above, the validation of the new scale and introducing it to the clinical practice would take much time which is critical in the pandemic setting.Noteworthy the CHA2DS2Vasc score, commonly used in clinical practice for estimating the risk of stroke in people with atrial fibrillation (AF), includes commorbidities such as diabetes but also congestive heart failure , hypertension, prior stroke/TIA or thromboembolism, vascular disease (e.g. peripheral artery disease, myocardial infarction, aortic plaque) and sex category. Similar to the C2HEST score, it is well validated and based on the simple analysis of comorbidities. We postulate that the CHA2DS2Vasc score might also have progrnostic value in predicting the COVID-19 outcome in elderly subjects, which requires further detailed and extensive analyses.
We fully agree with the suggestion made by the Reviewer regarding potential benefits from the expansion of the C2HEST scale-especially in the field of increased predictive ability –which is now more extensively discussed.
In order to address the Reviewers suggestion regarding including obesisty and diabetes mellitus to the extended version of the C2HEST score (C2HEST-OD), we have performed such analysis, which is presented below and in the supplementary material, as follows:
- C2HEST-OD – expansion of the C2HEST score by adding the presence of diabetes mellitus and obesity
In order to verify if including obesisty and diabetes mellitus to the extended version of the C2HEST score (C2HEST-OD), we have performed the analysis presented below:
A.1 All-cause mortality (Suppl. Table 2)
Suppl. Table 2
|
|
Hazard ratio |
Lower CI limit |
Upper CI limit |
p-value |
|
Overall |
1.2382 |
1.1335 |
1.3525 |
<0.0001 |
|
low vs medium |
2.6670 |
1.3568 |
5.2422 |
0.0044 |
|
low vs high |
4.2077 |
2.2037 |
8.0340 |
<0.0001 |
The change from the low to the high risk group increased the risk for death 4.21-fold vs 2.70 fold in the original version of the scale.
A.2 In-hospital mortality – suppl. Table 3
Suppl. Table 3
|
|
Hazard ratio |
Lower CI limit |
Upper CI limit |
p-value |
|
Overall |
1.2052 |
1.0821 |
1.3424 |
0.0007 |
|
low vs medium |
1.7960 |
0.8411 |
3.8348 |
0.1303 |
|
low vs high |
2.8402 |
1.3798 |
5.8462 |
0.0046 |
A.3. Clinical non-fatal events
The strength of the association between CH2EST-OD score and study endpoints including non-fatal adverse events is presented in supplementary Table 4
Supplementary Table 4.The strength of the association between CH2EST-score and study endpoints.
|
Endpoint |
Type |
Odds.ratio |
Lower.CI.limit |
Upper.CI.limit |
p.value |
|
end of hospitalization - full recovery |
Overall |
0.749 |
0.642 |
0.865 |
0.0001 |
|
|
low vs medium |
0.421 |
0.195 |
0.884 |
0.0244 |
|
|
low vs high |
0.219 |
0.101 |
0.456 |
<0.0001 |
|
end of hospitalization - deterioration |
Overall |
1.044 |
0.798 |
1.343 |
0.7451 |
|
|
low vs medium |
1.791 |
0.369 |
12.871 |
0.4968 |
|
|
low vs high |
2.154 |
0.496 |
14.858 |
0.3508 |
|
end of hospitalization - rehabilitation |
Overall |
1.059 |
0.854 |
1.303 |
0.5888 |
|
|
low vs medium |
0.969 |
0.291 |
3.454 |
0.9596 |
|
|
low vs high |
1.200 |
0.399 |
4.057 |
0.7529 |
|
end of hospitalization - death |
Overall |
1.290 |
1.120 |
1.497 |
0.0005 |
|
|
low vs medium |
2.575 |
1.113 |
6.402 |
0.0325 |
|
|
low vs high |
4.245 |
1.901 |
10.304 |
0.0007 |
|
all-causeshock |
Overall |
1.006 |
0.864 |
1.168 |
0.9337 |
|
|
low vs medium |
1.760 |
0.763 |
4.281 |
0.1952 |
|
|
low vs high |
1.397 |
0.611 |
3.367 |
0.4388 |
|
hypovolemicshock |
Overall |
0.668 |
0.403 |
1.000 |
0.0769 |
|
|
low vs medium |
1.169 |
0.273 |
5.922 |
0.8359 |
|
|
low vs high |
NA |
NA |
NA |
NA |
|
cardiogenicshock |
Overall |
1.726 |
1.183 |
2.664 |
0.0069 |
|
|
low vs medium |
NA |
NA |
NA |
NA |
|
|
low vs high |
NA |
NA |
NA |
NA |
|
septicshock |
Overall |
1.057 |
0.902 |
1.235 |
0.4897 |
|
|
low vs medium |
1.899 |
0.746 |
5.289 |
0.1935 |
|
|
low vs high |
2.145 |
0.877 |
5.826 |
0.1095 |
|
pulmonaryembolism |
Overall |
0.798 |
0.571 |
1.068 |
0.1551 |
|
|
low vs medium |
1.958 |
0.534 |
9.300 |
0.3396 |
|
|
low vs high |
0.378 |
0.048 |
2.355 |
0.2954 |
|
deepveinthrombosis |
Overall |
0.957 |
0.273 |
2.376 |
0.9291 |
|
|
low vs medium |
NA |
NA |
NA |
NA |
|
|
low vs high |
NA |
NA |
NA |
NA |
|
venousthromboembolicdisease |
Overall |
0.798 |
0.571 |
1.068 |
0.1551 |
|
|
low vs medium |
1.958 |
0.534 |
9.300 |
0.3396 |
|
|
low vs high |
0.378 |
0.048 |
2.355 |
0.2954 |
|
myocardialinfarction |
Overall |
1.202 |
0.842 |
1.694 |
0.2929 |
|
|
low vs medium |
0.338 |
0.015 |
3.623 |
0.3813 |
|
|
low vs high |
1.185 |
0.223 |
8.779 |
0.8478 |
|
myocardialinjury |
Overall |
1.169 |
0.991 |
1.389 |
0.0678 |
|
|
low vs medium |
2.059 |
0.758 |
6.063 |
0.1687 |
|
|
low vs high |
3.217 |
1.216 |
9.309 |
0.0229 |
|
acuteheartfailure |
Overall |
2.699 |
1.874 |
4.315 |
<0.0001 |
|
|
low vs medium |
NA |
NA |
NA |
NA |
|
|
low vs high |
NA |
NA |
NA |
NA |
|
stroke/TIA |
Overall |
0.413 |
0.025 |
1.458 |
0.3357 |
|
|
low vs medium |
NA |
NA |
NA |
NA |
|
|
low vs high |
NA |
NA |
NA |
NA |
|
complete respiratory failure |
Overall |
1.054 |
0.784 |
1.433 |
0.7266 |
|
|
low vs medium |
0.343 |
0.038 |
2.267 |
0.2867 |
|
|
low vs high |
0.550 |
0.067 |
3.319 |
0.5322 |
|
SIRS |
Overall |
1.072 |
0.865 |
1.318 |
0.5163 |
|
|
low vs medium |
1.238 |
0.353 |
4.955 |
0.7443 |
|
|
low vs high |
1.709 |
0.548 |
6.456 |
0.382 |
|
sepsis |
Overall |
1.155 |
0.593 |
2.164 |
0.6393 |
|
|
low vs medium |
NA |
NA |
NA |
NA |
|
|
low vs high |
NA |
NA |
NA |
NA |
|
acutekidneyinjury |
Overall |
1.179 |
1.015 |
1.373 |
0.0315 |
|
|
low vs medium |
1.899 |
0.746 |
5.289 |
0.1935 |
|
|
low vs high |
3.181 |
1.332 |
8.517 |
0.0133 |
|
acuteliverdysfunction |
Overall |
1.294 |
0.965 |
1.738 |
0.081 |
|
|
low vs medium |
2.104 |
0.260 |
43.258 |
0.5249 |
|
|
low vs high |
3.711 |
0.608 |
71.237 |
0.2315 |
|
MODS |
Overall |
1.092 |
0.787 |
1.483 |
0.5819 |
|
|
low vs medium |
2.130 |
0.264 |
43.765 |
0.5179 |
|
|
low vs high |
3.062 |
0.476 |
59.595 |
0.3134 |
|
allbleedings |
Overall |
1.058 |
0.830 |
1.334 |
0.6352 |
|
|
low vs medium |
3.871 |
0.963 |
25.955 |
0.0898 |
|
|
low vs high |
1.500 |
0.310 |
10.758 |
0.6358 |
|
intracranialbleeding |
Overall |
0.837 |
0.334 |
1.633 |
0.6408 |
|
|
low vs medium |
NA |
NA |
NA |
NA |
|
|
low vs high |
NA |
NA |
NA |
NA |
|
respiratory tractbleeding |
Overall |
1.244 |
0.851 |
1.802 |
0.2441 |
|
|
low vs medium |
NA |
NA |
NA |
NA |
|
|
low vs high |
NA |
NA |
NA |
NA |
|
upper-GI-tractbleeding |
Overall |
1.036 |
0.683 |
1.505 |
0.8572 |
|
|
low vs medium |
2.130 |
0.264 |
43.765 |
0.5179 |
|
|
low vs high |
1.181 |
0.110 |
25.795 |
0.8932 |
|
lower-GI-tract-bleeding |
Overall |
0.957 |
0.273 |
2.376 |
0.9291 |
|
|
low vs medium |
NA |
NA |
NA |
NA |
|
|
low vs high |
NA |
NA |
NA |
NA |
|
urinarytractbleeding |
Overall |
0.710 |
0.243 |
1.456 |
0.4264 |
|
|
low vs medium |
0.690 |
0.027 |
17.731 |
0.7949 |
|
|
low vs high |
NA |
NA |
NA |
NA |
|
pneumonia |
Overall |
1.058 |
0.920 |
1.220 |
0.4353 |
|
|
low vs medium |
0.938 |
0.440 |
1.978 |
0.8659 |
|
|
low vs high |
1.276 |
0.605 |
2.670 |
0.5175 |
|
new cognitive signs and symptoms |
Overall |
0.990 |
0.667 |
1.407 |
0.9577 |
|
|
low vs medium |
NA |
NA |
NA |
NA |
|
|
low vs high |
0.774 |
0.164 |
4.067 |
0.7439 |
|
Admissionat ICU |
Overall |
0.931 |
0.806 |
1.071 |
0.3214 |
|
|
low vs medium |
1.097 |
0.516 |
2.363 |
0.8103 |
|
|
low vs high |
0.954 |
0.457 |
2.019 |
0.8997 |
|
|
|
|
|
|
|
Suppl. Figure 2.
Ad.2 In order to clarifythe study inclusion criteria we rephrasedpart of Material and Methods section as follows:
“The soleinclusion criterion to this subanalysis was age of>65 years in the COVID-19 patients”.
Moreover, in order to be clearer and more precise in defining the study design, weadded the study-flow diagram to highlight the details of our study protocol:
Ad.3We believe that the Reviewer made a valid point while stating that considering the initial COVID severity and IC admission as well the C2HEST score sub-analyses would provide an important information for understanding its usefulness in predicting the COVID outcome.
As result, additional data has been added. In the Table 3 the analysis of the VES-13 (Vulnerable of Elderly Score) and the Glasgow Coma Scale are presented as follows:
|
VES-13, points mean+SD min-max N=75 |
Low 4.24+2.99 1-9 17 |
Medium 5.58+3.3 1-12 36 |
High 6.54+2.89 3-13 22 |
p value 0.067
|
post hoc p N/A
|
|
GCS, points mean+SD min-max N=402 |
14.57+1.75 3-15 133 |
14.38+1.81 3-15 160 |
14.18+2.27 3-15 109 |
0.305
|
N/A
|
and in the Discussion section:
Therefore, it is crucial to identify potentially severe cases and implement immediately effective treatment to prevent the progression of the disease from its beginning. Interestingly, there were no significant differences on admission in terms of the Vulnerable Elderly Score which is a simple scoring system capable of identifying vulnerable elderly people in the community and includes factors such as age, self-assessed health, functional limitations, and impairments [32]. Health vulnerability is associated with a higher risk of mortality and functional decline in older people in the community. However, few studies have evaluated the role of the Vulnerable Elders Survey (VES-13) in predicting clinical outcomes of hospitalized patients [33,34]. One of the recent studies, based on the small cohort (n=91) suggests that elderly patients (>60 years) classified as extremely vulnerable had more unfavorable outcomes after hospitalization for COVID-19 - super vulnerability was an independent predictor of death and the need for invasive mechanical ventilation during hospitalization - a final VES-13 score between 8 and 10 was associated with poor outcomes [35]. Our results show lack of significant differences in the VES-13 between the three C2HEST strata. Similarly, we did not observe differences in the GCS score between the risk strata, which could point thus at an independent predicting value of the C2HEST score in the fatal and non-fatal outcomes of elderly subjects with COVID-19. In the supplementary file we have presented the usefulness of the C2HEST score in elderly subjects who were admitted directly to the intensive care unit (due to COVID severity) vs. those admitted to the non-intensive ward of the medical university center due to COVID-19. The C2HEST score revealed to determine the outcome (mortality and non-fatal adverse clinical events) irrespectively on the initial symptom severity. Noteworthy, C2HEST score predicted also the mortality irrespectively on the tranfer to the ICU, which might point at its additional value in better predicting the need for advanced supportive care and performing better triage of subjects being at greater risk for death who could take an advantage of earlier escalation of the monitoring and supportive care.
Moreover, a sub-analysis of the survival was performed and additional supplementary data has been added:
- Sub-analysis of the survival based on the ICU admission in particular C2HEST score strata
The presented below Kaplan-Meier (suppl. Figure 3)survival function presents the usefulness of the C2HEST score in predicting the mortality of elderly subjects who were admitted directly to the intensive care unit (due to COVID severity) vs. those admitted to the non-intensive ward of the medical university center due to COVID-19. The C2HEST score revealed to determine the outcome irrespectively on the initial symptom severity with the log-rank p-value of <0.0001:
Suppl. Figure 3.
The C2HEST predictive value in transfer of COVID-19 elderly subjects to the ICU following clinical deterioration is presented in the Suppl. Table 5:
Suppl. Table 5:
|
Endpoint |
Type |
Odds.ratio |
Lower.CI.limit |
Upper.CI.limit |
p.value |
|
Admission at ICU |
Overall |
0.950 |
0.841 |
1.068 |
0.3958 |
|
|
low vs medium |
0.593 |
0.366 |
0.950 |
0.031 |
|
|
low vs high |
0.755 |
0.442 |
1.261 |
0.2913 |
Also, the logistic regression model was developed to analyze the description of the admission at ICU by particular C2HEST components, where the strongest predictive value for ICU admission had the presence of hypertension, coronary artery disease and HFrEF (supplementary Table 6).
Supplementary Table 6:
|
Endpoint |
Component |
Odds.ratio |
Lower.CI.limit |
Upper.CI.limit |
p.value |
|
Admission at ICU |
Coronary artery disease |
1.443 |
0.831 |
2.446 |
0.1814 |
|
|
COPD |
0.631 |
0.212 |
1.508 |
0.3476 |
|
|
Age |
0.351 |
0.218 |
0.549 |
<0.0001 |
|
|
Thyroid Disease |
0.373 |
0.142 |
0.810 |
0.0239 |
|
|
Hypertension |
1.794 |
1.092 |
3.051 |
0.0251 |
|
|
HFrEF |
1.519 |
0.879 |
2.574 |
0.1261 |

Reviewer 2 Report
Minor points:
modify the title in '':a subanalysis of the COLOS study.''
Introduction
- line 67, The pandemic required preventive measures for personal protection, through the use of hygiene and preventive rules in order to limit viral spread. please cite doi:10.7416/ai.2021.2439.
- line 72, Emerging search for patient factors is relevant to correctly selecting the patient, including demographic, clinical, immunological, hematological, biochemical, and radiographic findings, which may be useful to clinicians in predicting COVID-19 severity and mortality. It predicts factors such as age> 55 years, multiple pre-existing comorbidities, hypoxia, specific computed tomography findings indicative of extensive lung involvement, various laboratory test abnormalities, and biomarkers of organ dysfunction.please cite doi:10.1002/rmv.2146
Methods
statistical analysis interesting. Add a flow diagram to explain study protocol
Results
well written, excellent tables.
Discussion
- line 378, Older individuals are more susceptible to various infections due to the immunological changes that occur during the aging process. These changes collectively referred to as "immunosenescence" include decreases in innate and adaptive immune responses in addition to the exacerbated production of inflammatory cytokines. This immunological dysfunction scenario and its relationship to the development of the disease in the elderly has been extensively studied, especially in infections that can be fatal, such as influenza and, more recently, COVID-19., please cite doi:10.3389/fimmu.2020.579220
Author Response
We would like to thank the Reviewer for an in-depth analysis of the manuscript and for pivotal comments provided, which have resulted in a significant improvement of this manuscript.
Introduction section:
Ad 1 As the Reviewer suggested, we havecommented on the non-pharmacological methods focused on the limitation of viral spread and the reference regarding this issue have been added.
Ad 2 Another valid point mentioned by the Reviewer is a remark regarding, immunological, hematological, biochemical, and radiographic findings, which may be useful to clinicians in predicting COVID-19 severity and mortality. We have added necessary comment in manuscript along with proposed references and would like to thank the Reviewer for this suggestion.
Material and methods section:
Ad 1. We believe that the Reviewer made a valid point when suggested adding study-flow diagramto highlight the details of study protocol.
Discussion section
Ad 1. We believe that the Reviewer made a valid point when suggested adding comment regarding immunological changes in elderly population. Hence we added “ Especially the elder population with co-occurring immunological changes named collectively as "immunosenescence" [27] -connected with a decrease of innate and adaptive immune responses and exacerbation in the production of inflammatory cytokines - during the aging process is susceptible to various infections and requires careful initial assessment."

Round 2
Reviewer 1 Report
My major comments were appropraitely adressed. No further coments